# Autonomic responses to aerobic and resistance exercise in patients with chronic musculoskeletal pain: A systematic review

**Hironobu Uzawa**[1]*, **Kazuya Akiyama**[2], **Hiroto Furuyama**[2], **Shinta Takeuchi**[1], **Yusuke Nishida**[1,2]

1 Department of Physical Therapy, School of Health Sciences at Narita, International University of Health and Welfare, Narita, Chiba, Japan, 2 Rehabilitation Center, International University of Health and Welfare Narita hospital, Narita, Chiba, Japan

* h.uzawa@iuhw.ac.jp

**Data Availability Statement:** All relevant data are within the paper and its Supporting information files.

## Abstract

### Background

It is unknown whether patients with chronic musculoskeletal pain (CMP) show autonomic dysregulation after exercise, and the interventional effects of exercise on the autonomic dysregulation have not been elucidated. The objectives of this study were to reveal acute autonomic responses after aerobic and resistance exercises and the interventional effects of both exercises on autonomic dysregulation in patients with CMP.

### Methods

A systematic search using nine electronic databases was performed based on three key search terms: "chronic musculoskeletal pain," "autonomic nervous system," and "exercise." Data were extracted from measurements of the autonomic nervous system and pain.

### Results

We found a total of 1170 articles; 17 were finally included, incorporating 12 observational and five interventional studies. Although a comparator has not been specified, healthy controls were compared to patients with CMP in observational studies. Three of five interventional studies were pre-post study with healthy controls as a comparator or no controls. The other two interventional studies were randomized controlled trial with a different treatment e.g., stretching. There were four good, 10 fair, and three poor-quality articles. The total number of participants was 617, of which 551 were female. There was high heterogeneity among the five disease conditions and nine outcome measures. Following one-time exposure to aerobic and resistance exercises, abnormal autonomic responses (sympathetic activation and parasympathetic withdrawal), which were absent in healthy controls, were observed in patients with CMP. The effects of aerobic and resistance exercise as long-term interventions were unclear since we identified both positive effects and no change in the autonomic activities in patients with CMP.

**Funding:** Initials of the authors who received: HU Grant numbers: 21K19733 (Challenging Research, Explorator) The full name of each funder: received the JSPS Grants-in-Aid for Scientific Research URL is: https://kaken.nii.ac.jp/grant/KAKENHI-PROJECT-21K19733/ The funders had no role in study design, data collection and analysis, decision to publish, or preparation of the manuscript.

**Competing interests:** The authors have declared that no competing interests exist.

## Conclusions

This study indicates dysfunctional autonomic responses following one-time exposure to exercise and inconsistent interventional effects in the autonomic activities in patients with CMP. Appropriate therapeutic dose is necessary for studying the management of autonomic regulation and pain after exercise.

## Introduction

Chronic musculoskeletal pain (CMP) has a negative impact on the physical activity, quality of life, and the social care system [1]. Exercise therapy is effective in alleviating CMP [2], which may be attributed to exercise-induced hypoalgesia [3]. Exercise-induced hypoalgesia indicates an increase in the pain threshold after a single bout of exercise that occurs in healthy individuals [4] and patients with chronic pain [5]. However, some cases of patients with chronic pain, such as fibromyalgia or chronic fatigue syndrome, exhibit decreased pain threshold after exercise, that is, deteriorated pain sensitivity [6]. Pain after exercise should be managed since it could result in fear of pain and consequently physical inactivity [7].

Exercise-induced hypoalgesia and its dysfunction are related to the autonomic nervous system since autonomic activities control the descending pain pathways, which are located in the central nervous system and activate this hypoalgesia [8]. An underlying mechanism of this relationship is that both the systems have the same neuron named ventromedial rostral medulla, which modulates peripheral nociception [9]. Furthermore, exercise-induced hypoalgesia is also controlled by the autonomic nervous system and hypothalamic-pituitary-adrenal axis, which releases anti-inflammatory and analgesic hormone such as cortisol [10]. However, chronic stress, such as chronic pain, deteriorates these systems [10]. For example, chronic pain violates the ventromedial rostral medulla resulting in dysfunction [11]. It also leads to the prolonged release of cortisol and decreases the sensitivity of cortisol receptors [12]. Therefore, abnormal exercise-induced hypoalgesia is caused by dysfunction of the autonomic nervous system.

On the other hand, it is unclear whether patients with CMP have autonomic dysfunctions. Oura et al. [13] conducted a birth cohort study recruiting 4186 people and concluded that there was no relationship between pain intensity and cardiovascular autonomic function. Generaal et al. [14] reported similar results in that dysregulated autonomic and stress systems were not associated with the onset of CMP. Although these articles did not accept a relationship between pain and autonomic dysfunction, some articles supported this relationship. Significant correlations between chronic neck-shoulder pain and autonomic dysfunction [15]. Patients with fibromyalgia also tend to exhibit autonomic dysregulation [16]. Furthermore, a recent article focused on patients with osteoarthritis and autonomic function [17]. As discussed above, whether patients with CMP have autonomic dysregulation is controversial. If this relationship was clarified, therapists could identify what kind of patients with CMP have a dysfunctional autonomic nervous system and, possibly, dysregulated exercise-induced hypoalgesia. Moreover, since exercise can improve the autonomic nervous system in healthy people [18] and patients with cardiovascular disease [19], it could be effective in improving the dysfunctional autonomic nervous system in patients with CMP. However, to our knowledge, there is insufficient evidence on the interventional effect of exercise on the autonomic nervous system of CMP patients.

This study aimed to examine whether patients with CMP show autonomic dysregulation after one-time exposure to aerobic and resistance exercise. Furthermore, this study also aimed

to elucidate whether prolonged use of aerobic and resistance exercise over many months is effective in alleviating the patients' autonomic activities. We conducted a systematic literature review to broadly and transparently collect and integrate the current research. The search results of this review could help therapists identify the cause of the dysfunction in exercise-induced hypoalgesia and manage the patients' pain.

## Methods

### Registration and deviations from protocol

This systematic review was performed in accordance with the guidelines for systematic reviews of musculoskeletal diseases [20] and the Preferred Reporting Items for Systematic Reviews and Meta-Analysis (PRISMA) statement [21]. PRISMA checklist is shown in (S1 Table). This review was registered in PROSPERO on 30th August 2021 (ID: CRD42021265767). The protocol of this systematic review was originally to perform a meta-analysis of data on the autonomic nervous system; however, a meta-analysis could not be conducted because of different populations and outcome measures. Therefore, the title and protocol of PROSPERO were changed on 13th March 13, 2022, and the Synthesis Without meta-analysis framework was utilized as a guideline for data synthesis [22]. Heterogeneity was not calculated.

### PICO/PECOs and eligibility criteria

The PICO/PECOs for this review are as follows: P, patients with CMP; I/E, aerobic and resistance exercise; C, not specified; O, autonomic nervous system. The study types were observational, interventional, and case studies. The eligibility criteria were defined before the systematic search based on the PICO/PECOs and the aims of this study. The eligibility criteria are summarized in Table 1.

Patients with CMP were included in the study. CMP was defined as pain lasting ≥3 months in any of the following body parts: neck, lower back, shoulder, elbow, hand, wrist, hip, knee, ankle, or foot [23,24]. The intervention/exposure was aerobic and resistance exercise. Aerobic exercise was defined as any physical activity that was maintained continuously and rhythmically, and supplied adenosine triphosphate by aerobic metabolism (e.g., bicycle ergometer, treadmill, arm ergometry) [25]. Resistance exercise is defined as intense muscle contractions with repetitions to develop power and strength [26]. It is divided into isometric and dynamic exercises according to a previous systematic review [6]. The comparator was not specified because this study aimed to identify the effect of aerobic and resistance exercises on autonomic variables regardless of the type of comparator. The outcome was a measure of the autonomic

**Table 1. Eligibility criteria.**

| PICO/PECOs | Inclusion criteria | Exclusion criteria |
|---|---|---|
| Patients | Patients with chronic musculoskeletal pain | Any other body parts with pain, cancer-related pain, visceral pain, Animal studies |
| Interventions/ Exposures | Aerobic and resistance exercise | Stretching, yoga, breathing exercises combined program |
| Comparisons | Not specified | Not specified |
| Outcomes | Autonomic nervous system | Not specified |
| Type of study | Interventional, observational, and case studies | Not specified |

This table describes the eligibility criteria in accordance with PICO/PECOs.

**Table 2. Outcome measures for the autonomic nervous system.**

| Outcome measures | Explanation |
|---|---|
| Heart rate variability | Analysis of beat-to-beat fluctuations that indicate the sympathetic and parasympathetic nervous systems. Frequency (e.g., HF and LF/HF) and time (e.g., RMSSD) domain analysis. |
| Heart rate recovery | A difference between peak HR during exercise and HR after exercise. HRR showing an effect of autonomic adjustment |
| Chronotropic reserve | The formula is as follows: (peak heart rate—resting heart rate/220-age-resting heart rate)×100 It indicates abnormal increases in the heart rate. |
| Heart rate | The speed of the heart, beat-to-beat. |
| Baroreflex sensitivity/control | The baroreceptor reflex system controls the variability of blood pressure. Its assessment is named "baroreflex sensitivity" for the evaluation of autonomic control of the cardiovascular system. It is expressed as baroreflex sensitivity or control. |
| Muscle sympathetic nervous activity | The electrical activity of the postganglionic sympathetic nervous system during muscle contractions. |
| Blood pressure | Pressure of the blood circulation. |
| Blood sample | Evaluation of autonomic neuroendocrine concentration. Adrenaline, Noradrenaline, Adrenocorticotropic hormone, and cortisol are examples of measures. |

HF, high frequency; HR, heart rate; HRR, heart rate recovery; LF, low frequency; RMSSD, root mean square of successive differences between normal heartbeats.

nervous system, which can be evaluated using different methods [27]. Table 2 presents a list of the outcome measures for the autonomic nervous system.

This systematic review included the following types of study: interventional, observational, and case study. Interventional studies were divided into randomized controlled trials (RCT), non-randomized controlled trials (non-RCTs), and pre-post studies [28]. A pre-post study examined outcomes before and after a particular intervention with temporality [28]. Although RCT compared two interventions in the same cohort, a pre-post study explored the differences between multiple cohorts (e.g., a particular patient group vs. healthy control) [28]. An observational study was a cross-sectional study that assessed outcomes before and after exposure [28]. Articles written in English were eligible for this systematic review. The publication date was not limited.

## Information sources and search strategies

A literature search was conducted between 8th-19th September, 2021. Nine electronic databases were used: MEDLINE, EMBASE, CINAHL, PsycINFO, ScienceDirect, Cochrane Central Register of Controlled Trials, PEDro, ClinicalTrials.gov, and the World Health Organization International Clinical Trial Registry Platform portal. The search strategy was based on the following three key terms: "chronic musculoskeletal pain," "autonomic nervous system," and "exercise." This search strategy was defined and uploaded to PROSPERO before the literature search. The full search strategy for MEDLINE is presented in (S2 Table). Relevant systematic reviews and clinical trials were manually reviewed to identify articles in accordance with the eligibility criteria. The results of the literature searches were merged and saved in EndNote X8 (Thomson Reuters, New York, U.S.). These systematic literature searches were independently undertaken by two authors (HU and KA), and any conflicts were discussed and resolved by a third author (HF).

## Study selection

Duplicate records of the same article were removed using EndNote X8 (Thomson Reuters, New York, NY, U.S.). The titles and abstracts of the remaining articles were independently screened by HU and KA to exclude unrelated articles. The full text of the screened articles was independently examined by the same authors for compliance with the inclusion and exclusion criteria. Any discrepancies between authors HU and KA were discussed, and the final decision was made by the third author (HF). The PRISMA flow diagram was used to illustrate the overall study selection [21].

## Data extraction and synthesis

Data from the included studies were independently extracted by authors HU and FH, and conflicts were resolved by the third author (KA) via discussion. The extracted data included author name, publication year, study design, exercise mode, participant characteristics (disease condition, sample size, age, and sex), exercise mode (aerobic or resistance exercise), intervention/exposure and control conditions, and findings regarding the autonomic nervous system and pain variables. The data were tabulated using an electronic spreadsheet (Microsoft, Washington, U.S.).

The findings of the included studies were grouped by study design as observational or interventional studies. The reason for grouping was that this review had two aims. An observational study would be better suited for exploring whether patients with CMP showed autonomic dysregulation after one-time exposure to aerobic and resistance exercise [28]. The effects of therapeutic aerobic and resistance exercises over many months would be better evaluated using the findings of interventional studies.

The primary outcome was the measurement of the autonomic nervous system and the secondary outcome was pain variables. They were described using a visual analog scale, pressure pain threshold, and specific questionnaires. The primary and secondary outcomes were extracted using standardized metrics (mean differences, odds ratios, and risk ratios) and certainty of evidence (confidence intervals and p-values) [22]. Heterogeneity was described in an evidence map introduced by Foulds [29]. It is effective in graphically showing the heterogeneity between different study designs, populations, and outcome measures. The data are presented as tables and harvest plots. Harvest plots were used to illustrate the positive or negative findings of each included study with the result of quality assessment of bias, which was beneficial when meta-analysis could not be performed because of study diversity [30].

## Quality assessment

The methodological quality of the included studies was assessed using the Downs and Black checklist [31], as recommended in the guidelines [20]. The checklist can be used to assess randomized and non-randomized controlled trials in terms of reporting, internal and external validity, and power [31]. Total scores of 28–24, 23–19, 18–13, and ≤12 were graded as excellent, good, fair, and poor, respectively [32]. KA and FH independently assessed methodological quality, and HU joined a discussion if disagreements existed. Agreement of the quality assessments between examiners was estimated using Cohen's Kappa coefficient, where the Kappa values were graded as 1.0–0.81 = very good agreement, 0.80–0.61 = good agreement, 0.60–0.41 = moderate agreement, and < 0.4 = poor agreement [33].

## Results

### Results of study selection

Nine electronic databases were searched, and 1170 articles were identified. A manual search for relevant systematic reviews and clinical trials did not reveal any articles. After removing duplicates, titles and abstracts of 1029 articles were screened. We reviewed the full texts of 42 articles to assess their eligibility, and 17 articles were finally included. A PRISMA flow diagram is shown in Fig 1.

### Description of included studies

Twelve observational studies and five interventional studies were included. Tables 3 and 4 show the detailed descriptions of the included observational and interventional studies, respectively. All the included observational studies were cross-sectional studies comparing before and after exercise. Of the five interventional studies, two were RCT, and the remaining articles were pre-post trials. No case has been reported to date. Eight and nine articles reported aerobic and resistance exercises, respectively. The total number of participants in the included articles was 617, of which 551 were female. The disease conditions included fibromyalgia, chronic fatigue syndrome, ankylosing spondylitis, rheumatoid arthritis, and chronic neck and shoulder pain. The outcome measures included heart rate variability, heart rate recovery, chronotropic reserve, heart rate, baroreflex sensitivity, muscle sympathetic nervous activity (MSNA), blood pressure, blood sample, and pain. An evidence map of heterogeneity among the included articles is shown in Fig 2.

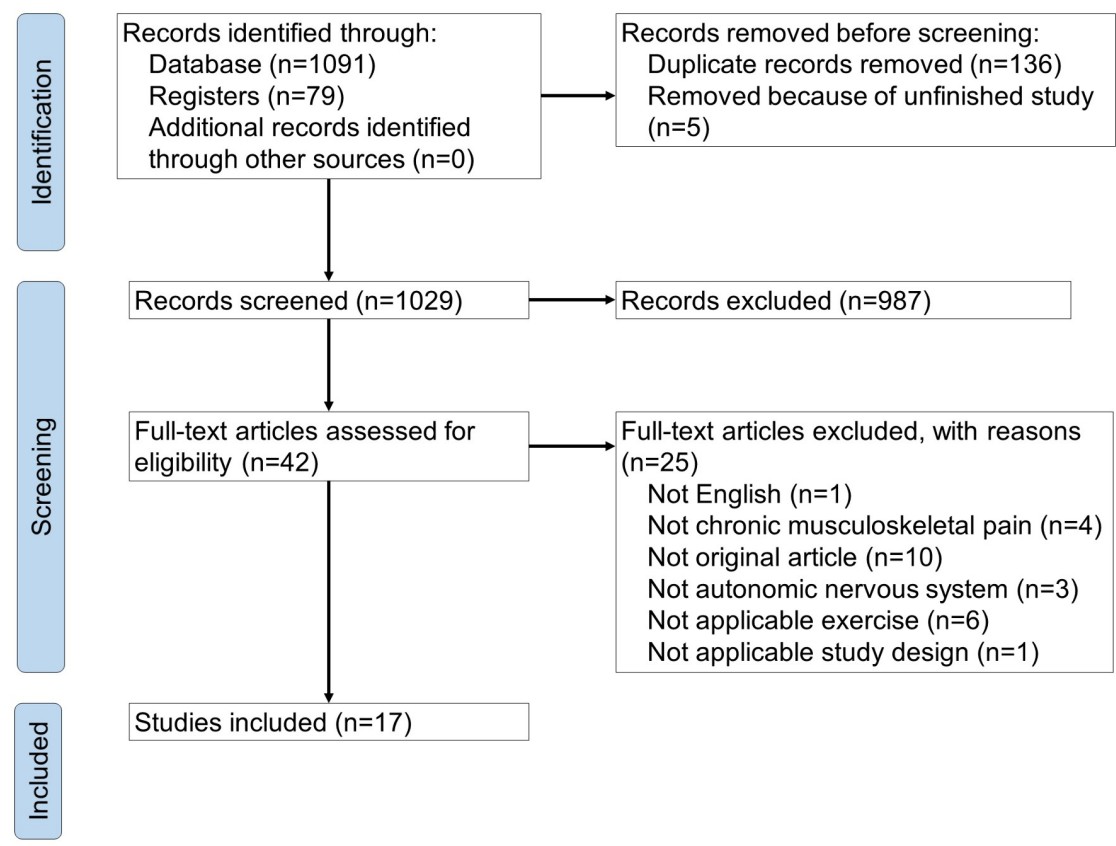

**Fig 1. PRISMA flow diagram of study selection.**

**Table 3. Description of included observational studies.**

| No | First Author (Year) | Study Design | Exercise Mode | Participants characteristic | Exposure | Protocol | Outcomes | Measurement timeframe |
|---|---|---|---|---|---|---|---|---|
| 1 | Cordero (1996) [48] | Cross-sectional | Aerobic | CFS 11 M1 F10, Ynone Healthy 11 M1 F10, Ynone | Treadmill Slow walking (2.5 mph) | 4 rest periods were alternated with 4 walking periods. A recovery period followed the last walk period | HRV, HR | Rest, walking, and recovery periods |
| 2 | da Cunha Ribeiro (2011) [49] | Cross-sectional | Aerobic | FM 14 M0 F14, Y46±3 Healthy 14 M0 F14, Y41±4 | Treadmill Maximal exercise test | Treadmill incremental test with a 2-minute recovery period | HRR, CR HR | The first and second minutes in the recovery period. |
| 3 | Giske (2008) [38] | Cross-sectional | Resistance | FM 19 M0 F19, Y37±7 Healthy 19 M0 F19, Y37±7 | Repetitive isometric contractions | 6-second quadriceps contraction (30% MVC) with 4-second rest until exhaustion (cannot achieve 30% MVC) After contraction, 30-minute recovery periods are followed by 30 minutes of rest. | HR, BS, Pain | Baseline, minutes 1 and 6 in the exercise period, and 1, 5, 10, 15, 20, and 30 minutes during the recovery period |
| 4 | Kadetoff (2010) [39] | Cross-sectional | Resistance | FM 16 M0 F16, Y38.2 (ranged 22–56) Healthy 16 M0 F16, Y38.3 (ranged 22–53) | Isometric contraction | Following 30-minute rest, isometric contraction with quadriceps femoris until exhaustion. | HR, BP, BS, Pain | Baseline, after 2.5 minutes of contraction, time at exhaustion, and after 30 minutes of recovery |
| 5 | Kadetoff (2007) [40] | Cross-sectional | Resistance | FM 17 M0 F17, Y38.8 (ranged 22–56) Healthy 17 M0 F17, Y37.4 (ranged 22–53) | Isometric contraction | Continuous isometric knee contraction until a maximum of 15 minutes or exhaustion with recovery periods. | HR, BRC, BP, Pain | Baseline before contraction, and 0, 5, 10, and 15 minutes after contraction. |
| 6 | Kaya (2010) [41] | Cross-sectional | Aerobic | AS 28 M24 F4, Y28.7±5.7 Healthy 30 M26 F4, Y29.3±5.8 | Treadmill exercise testing | Treadmill until reaching 85% of maximal HR | HRV, HRR | Recovery period at 1, 2, and 3 minutes after treadmill |
| 7 | Kingsley (2009) [42] | Cross-sectional | Resistance | FM 9 M0 F9, Y48 ±4 Healthy 9 M0 F9, Y48±2 | whole body resistance exercise | After 20 minutes of rest, 10 resistance exercises for 30 minutes. 20 minutes recovery period following exercise | HRV, HR, BRS, Pain | 15 minutes after rest and recovery periods started |
| 8 | Maia (2016) [43] | Cross-sectional | Aerobic | FM 25 M7 F18, Y15 (ranged 11–17) Healthy 25 M7 F18, Y15 (ranged 11–17) | Incremental exercise using a treadmill | Treadmill walking protocol with incremental speed every 1 minute. Two minutes of recovery period following exercise. | HRR, CR, HR | During exercise and 1 and 2 minutes after exercise. |
| 9 | Oosterwijck (2017) [44] | Cross-sectional | Aerobic | CFS 20 M0 F20, Y41.6±9.8 Healthy 20 M0 F20, Y34.6 ±15.2 | Bicycle ergometer | 25W/ minute incremental bicycle ergometer until 75% of maximal HR with a subsequent recovery period. | HRV, HR, BP, Pain | Pre-exercise and post-exercise after 10 minutes |
| 10 | Peçanha (2021) [34] | Cross-sectional | Resistance | RA 33 M0 F33, Y61±7 Healthy 10 M0 F10, Y61±6 | Isometric contraction | After a 15-minute rest, isometric knee contraction for 3 minutes. | HRV, BRS, MSNA | During exercise. |
| 11 | Peçanha (2018) [45] | Cross-sectional | Aerobic | RA 27 M0 F27, Y59.3±5.8 Healthy 14 M0 F14, Y56.1±5.4 | Incremental exercise using a treadmill | Treadmill walking with increased inclination and speed every minute and during the recovery period. | HRR, CR, HR | Before and after exercise, recovery period 0.5, 1, 2, and 3 minutes after exercise |

(*Continued*)

**Table 3.** (Continued)

| No | First Author (Year) | Study Design | Exercise Mode | Participants characteristic | Exposure | Protocol | Outcomes | Measurement timeframe |
|---|---|---|---|---|---|---|---|---|
| 12 | Shiro (2012) [50] | Cross-sectional | Resistance | chronic neck and shoulder pain 14 M0 F14, Y29.5±4.1 Healthy 12 M0 F12, Y28.7±4.6 | Isometric contraction | Alternative three isometric upper trapezius contraction with 2 kg weight for 1 minute, and recovery periods for 2 minutes. | HRV | Pre-rest, three contractions, three recovery periods, and post-rest. |

AS, ankylosing spondylitis; BP, blood pressure; BRC, baroreflex control; BRS, baroreflex sensitivity; BS, blood samples; CFS, chronic fatigue syndrome; CR, chronotropic reserve; F, female; FM, fibromyalgia; HR, heart rate; HRR, heart rate recovery; HRV, heart rate variability; M, male; MSNA, muscle sympathetic nervous activity; RA, rheumatoid arthritis; Y, years old.

## Quality assessment of included studies

The risk of bias is summarized in Table 5, and the precise results are available in (S3 Table). The number of excellent articles was zero, four [34–37] were good, 10 [38–47] were fair, and three [48–50] were poor. The kappa values of the examiners' correlations (KA and HF) indicated a very good agreement at 0.873.

**Table 4. Description of included interventional studies.**

| No | First Author (Year) | Study Design | Exercise Mode | Participants characteristic | Intervention | Control | Protocol | Outcome | Measurement timeframe |
|---|---|---|---|---|---|---|---|---|---|
| 13 | Bardal (2015) [46] | Pre-post study | Aerobic | FM 25 M0 F25, Y54 ±7.3 Healthy 25 M0 F25, Y52 ±8.8 | Incremental cycling test | None | Intervention: 3–4 intervals of supervised cycle ergometer twice a week for 12 weeks with an intensity of 50–75% of maximal oxygen consumption. | HRV, HRR, HR, BP, Pain | Before and after the intervention. |
| 14 | Figueroa (2008) [47] | Pre-post study | Resistance | FM 10 M0 F10, Y49±8 Healthy 9 M0 F9, Y50±10 | Resistance exercise using machines | None | Intervention: nine resistance exercises using machines at 50–80% of 1-repetition maximum test. Twice a week for 16 weeks. | HRV | Before and after the intervention. |
| 15 | Gavi (2014) [35] | RCT | Resistance | FM 66 Strengthening 35 M0 F35, Y:44.34 ±7.94 Flexibility 31 M0 F31, Y:48.65 ±7.60 | Strengthening exercise | Flexibility exercise | Strengthening: twice per week for 16 weeks. Resistance training at over 45% of 1- repetition maximum test using machines. Flexibility: stretching program in standing position. | HRV, Pain | Before and 1, 2, 3, and 4 months after the intervention. |
| 16 | Kingsley (2010) [36] | Pre-post study | Resistance | FM 9 M0 F9, Y:42±5 Healthy 15 M0 F15, Y45±5 | Whole body resistance exercise | None | Intervention: supervised resistance training twice a week for 12 weeks. 50–85% 1-repetition maximum test. | HRV, HR, Pain | Before and after the intervention. |
| 17 | Sañudo (2015) [37] | RCT | Aerobic | FM 28 Exercise 16 M0 F16, Y58±2 Control 12 M0 F12, Y55±2 | Treadmill | No intervention (maintain a normal lifestyle) | Intervention: walking program twice a week for 24 weeks targeting 60–65% and 75–80% of predicted maximum heart rate Control: no intervention (continuing normal daily life). | HRV, Pain | Before and after the intervention. |

AS, ankylosing spondylitis; BP, blood pressure; BRS, baroreflex sensitivity; BS, blood samples; CFS, chronic fatigue syndrome; CR, chronotropic reserve; F, female; FM, fibromyalgia; HR, heart rate; HRR, heart rate recovery; HRV, heart rate variability; M, male; MSNA, muscle sympathetic nervous activity; RA, rheumatoid arthritis; RCT, randomized controlled trial; Y, years old.

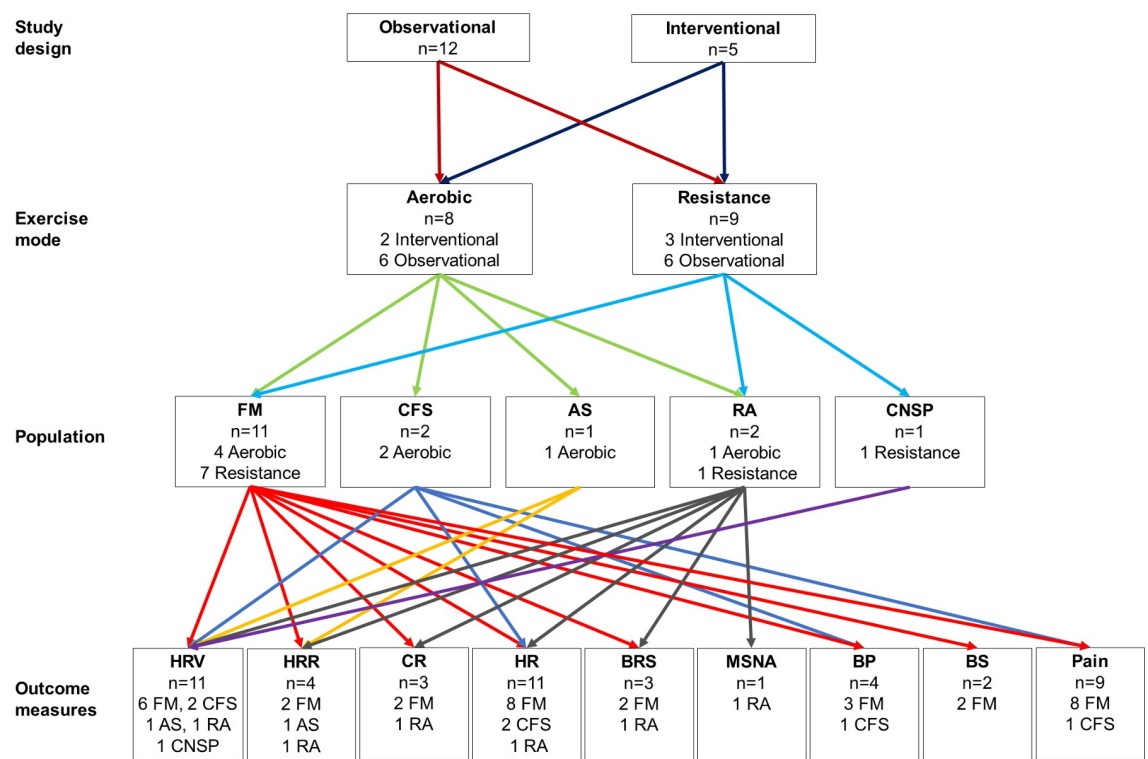

**Fig 2. Evidence map of the included studies.** This map shows the diversity of study design, exercise mode, population, and outcome measures.

## One-time exposure to aerobic or resistance exercise

Table 6 and Fig 3 show a summary of the main findings from the included observational studies. Heart rate variability after aerobic exercise was different in CMP patients from that in healthy controls. The low/ high frequency (LF/HF) ratio, which indicates sympathetic activity [51], was higher in CMP patients than in healthy controls [41,44] during exercise. Even after exercise, higher sympathetic activity tended to be prolonged [44]. Furthermore, the root mean square of successive differences between normal heartbeats (RMSSD) [41,44] and the percentage of adjacent normal-to-normal intervals (pNN50) [41] were lower in patients with CMP than in healthy controls. Both outcomes represent parasympathetic nervous system activity [51]. The HF also reflects parasympathetic neurons [51], and decreased HF occurs after aerobic exercise [44]. Cordero [48] showed decreased vagal power as walking sessions proceeded and at the recovery phase, although precise outcome measures were not included. Four articles [41,43,45,49] utilized heart rate recovery for three min after aerobic exercise. All articles showed significantly lower heart rate recovery in CMP patients than in healthy controls. Chronotropic reserve was assessed in three articles [43,45,49], and all showed lower chronotropic reserves in CMP patients than in controls. Although three articles [43,45,49] showed deteriorated responses in heart rate, two articles [44,48] did not show any differences. Blood pressure after aerobic exercise did not differ between groups [44]. Only one article [44] utilized pain as an outcome, and higher pain intensity in the patient group than in healthy controls was apparent after exercise.

In resistance exercise, three articles utilized heart rate variability [34,42,50], and two of the three articles [42,50] showed significant differences between CMP patients and healthy

**Table 5. Risk of bias assessment for included studies.**

| No | Author (Year) | Reporting (11) | External Validity (3) | Bias (7) | Confounding (7) | Power (1) | Total (28) | Grade |
|---|---|---|---|---|---|---|---|---|
| 1 | Cordero (1996) [48] | 6 | 0 | 3 | 1 | 0 | 10 | poor |
| 2 | da Cunha Ribeiro (2011) [49] | 8 | 0 | 3 | 1 | 0 | 12 | poor |
| 3 | Giske (2008) [38] | 9 | 1 | 4 | 2 | 1 | 17 | fair |
| 4 | Kadetoff (2010) [39] | 7 | 2 | 3 | 1 | 0 | 13 | fair |
| 5 | Kadetoff (2007) [40] | 7 | 2 | 3 | 1 | 0 | 13 | fair |
| 6 | Kaya (2010) [41] | 9 | 0 | 3 | 1 | 0 | 13 | fair |
| 7 | Kingsley (2009) [42] | 8 | 1 | 3 | 1 | 0 | 13 | fair |
| 8 | Maia (2016) [43] | 8 | 3 | 4 | 2 | 0 | 17 | fair |
| 9 | Oosterwijck (2017) [44] | 9 | 1 | 5 | 2 | 1 | 18 | fair |
| 10 | Peçanha (2021) [34] | 10 | 2 | 4 | 3 | 0 | 19 | good |
| 11 | Peçanha (2018) [45] | 9 | 2 | 3 | 2 | 0 | 16 | fair |
| 12 | Shiro (2012) [50] | 8 | 0 | 3 | 1 | 0 | 12 | poor |
| 13 | Bardal (2015) [46] | 11 | 2 | 3 | 2 | 0 | 18 | fair |
| 14 | Figueroa (2008) [47] | 10 | 1 | 3 | 3 | 0 | 17 | fair |
| 15 | Gavi (2014) [35] | 10 | 3 | 4 | 4 | 1 | 22 | good |
| 16 | Kingsley (2010) [36] | 9 | 2 | 5 | 3 | 0 | 19 | good |
| 17 | Sañudo (2015) [37] | 10 | 2 | 5 | 4 | 1 | 22 | good |

controls. LFnu/HFnu and LF in the patient group decreased post-exercise despite an increase in the healthy control group [42]. LF/HF in patients did not change during and after contraction but showed an increase in the healthy control group [50]. The HF also showed a different trend between the groups; it increased in patients and decreased in controls [42]. One of the three articles using heart rate variability [34] showed no group difference between patients and healthy controls in RMSSD. Three studies [34,40,42] utilized baroreflex sensitivity, and all three showed significant differences between the groups. Baroreflex sensitivity was lower in patients than in controls at baseline [34] and higher than that in healthy controls during [40,42] and after contraction [40]. Muscle sympathetic nervous activity was assessed in one study [34], and was found to be higher in pain patients than in healthy controls during and after resistance exercise. Blood samples were analyzed in two articles [38,39]; plasma adrenaline levels were lower in musculoskeletal pain patients than in healthy controls during exercise [38,39], and plasma noradrenaline levels increased during exercise, but there was no difference between patients and controls [38,39]. Similarly, plasma ACTH and cortisol did not show any change or group differences [38,39]. There were no group differences between patients and controls in terms of heart rate [38–40,42] or blood pressure [39,40]. Four articles [38–40,42] assessed pain during and after resistance exercise. Although three of the four articles [38–40] showed higher pain intensity in patients than in healthy controls, one article [42] did not show any difference.

## Effects of therapeutic interventions

Table 7 and Fig 4 show the main findings of the included interventional studies. Two papers [37,46] reported aerobic exercise as the intervention. One paper [37] reported statistical

**Table 6. Findings of observational studies.**

| No | Author (Year) | Summary of main findings |
|---|---|---|
| 1 | Cordero (1996) [48] | Vagal power was not different at baseline, but the patient group showed less vagal power than the control group during all walking and recovery periods (p<0.05). HR did not differ between the groups at baseline or walking sessions. |
| 2 | da Cunha Ribeiro (2011) [49] | Lower HRR (p = 0.002), CR (p<0.002), and HR (p<0.001) were found in the patient group than in the control group. |
| 3 | Giske (2008) [38] | Both groups showed an increased level of adrenaline during exercise (p<0.001). Peak adrenaline was lower in the patient group than in the control group (p = 0.03) Plasma noradrenaline and maximal cortisol showed no group difference (p = 0.29 and 0.19 respectively) HR did not differ between groups (p = 0.07) Higher pain observed in the patient group than in the control group during exercise (p = 0.004) |
| 4 | Kadetoff (2010) [39] | HR and BP were elevated during exercise, and no difference between the groups in HR and BP was found (p = 0.34 and 0.63 respectively) Plasma adrenaline was lower over time in patients than in controls (p<0.04), and both groups showed an increase compared with baseline (p<0.002). Plasma noradrenaline increased over time in both groups (p<0.0001), but no difference between the groups was found (p = 0.08). Plasma ACTH in patients did not show a statistical increase during exercise (p<0.23), and was lower at exhaustion (p<0.02). P-cortisol did not show any change or group difference (p<0.81 and p<0.92). Higher pain intensity in the patient group than in the control group was shown at all time points (p<0.001), but there was no increase before and after exercise in either group. |
| 5 | Kadetoff (2007) [40] | BRC was higher in patients than in control subjects at 2 minutes contraction and exhaustion (p<0.007 and 0.003 respectively) BP and HR increased during contraction and decreased at relaxation (p = 0.0001) without group differences (p = 0.24 in SBP and 0.86 in DBP). Pain (pressure pain threshold) in patients was higher than in controls at 2 minutes of contraction, exhaustion, 0, 5, and 10 minutes after contraction (p<0.005, 0.001, 0.001, 0.005 and 0.05 respectively). |
| 6 | Kaya (2010) [41] | LF and LF/HF were higher and SDNN, SDANN, RMSSD, and pNN50 were lower in patients than in controls (p<0.05), however, HF was not different (p>0.05) HRR at 1 and 2 minutes was lower in patients than in controls (p = 0.001), but HRR at 3 minutes was not different (p>0.05) |
| 7 | Kingsley (2009) [42] | LFnu and LFnu/HFnu in decreased post-exercise in patients, but increased in controls (p<0.05). HFnu increased in patients and decreased in controls (p<0.05). HR increased in both groups after exercise (p<0.05), and there were no group differences at rest and recovery periods (p = 0.14). BRS showed a significant group difference with an increase in patients and a decrease in controls (p<0.05). Pain (numeric rating scale) between the groups showed no differences. |
| 8 | Maia (2016) [43] | HRR at 1 and 2 minutes, CR, and peak HR were lower in patients than in controls. |
| 9 | Oosterwijck (2017) [44] | LF and HF at pre- and post-exercise were different between the groups (p<0.05), and LF/HF and RMSSD showed group differences only at post-exercise (p<0.05). RMSSD after exercise decreased in patients (p = 0.059) and increased in controls (p = 0.881) LF/HF after exercise increased in patients (p = 0.841) and decreased in controls (p = 0.502). LF and HF after exercise decreased in patients (p = 0.126 and 0.012 respectively) and controls (p = 0.044 and 0.709 respectively). HR and BP did not show any group differences at pre- and post-exercise. Higher pain intensity (visual analog scale) was seen in the patient group than in the control group at pre- and post-exercise (p<0.001). |

*(Continued)*

**Table 6.** (Continued)

| No | Author (Year) | Summary of main findings |
|---|---|---|
| 10 | Peçanha (2021) [34] | MSNA was higher in patients than in controls at baseline (p = 0.03). BRS was lower in patients at baseline (p = 0.06).*MSNA during and after exercise increased in patients (p<0.0001) and showed a higher value than in control (p = 0.04). HR and RMSSD were not different between the groups at baseline and after exercise. |
| 11 | Peçanha (2018) [45] | Lower HRR was seen in patients at 0.5, 1, 2, and 3 minutes compared with controls (p<0.05). CR in patients decreased, and CR in controls increased (p = 0.02). HR max was lower in patients than in controls (p = 0.005). |
| 12 | Shiro (2012) [50] | LF/HF increased in controls at contraction phases (p<0.01), but patients did not show significant change. LF/HF was lower in patients than in controls during contraction and recovery phases (p<0.05). |

BP, blood pressure; BRC, baroreflex control; BRS, baroreflex sensitivity; BS, blood samples; CR, chronotropic reserve; HF, high frequency; HR, heart rate; HRR, heart rate recovery; HRV, heart rate variability; LF, low frequency; MSNA, muscle sympathetic nervous activity; nu, normalized unit; RMSSD, root mean square of successive differences between normal heartbeats.

differences in the natural logarithm HF (LnHF) and LnLF/HF between patients and controls. Aerobic exercise also improved some measures of autonomic variables in the patient group [37]. However, another study [46] did not demonstrate any interventional effects in the standard deviations of all the normal-to-normal intervals (SDNN), RMSSD, or LF/HF. Similarly, heart rate recovery [46], heart rate [46], and blood pressure [46] were not significantly different between the groups. Bardal et al. [46] did not report an improvement in the overall pain. Sañudo et al. [37] did not also show statistical improvement in the pain of patients with fibromyalgia, while a trend of improvement was observed in the 10-cm visual analog scale for pain in patients from 7.4 ± 2.2 at baseline to 6.7 ± 2.2 following the intervention period.

Resistance exercise was utilized in three studies [35,36,47], and all of them evaluated heart rate variability. Figueroa et al. [47] showed interventional effects in the total power (TP), HF, and RMSSD; however, there were no group comparisons. Although Gavi et al. [35] showed only an improvement at pNN50, other measures at TP, RMSSD, LF, HF, LFnu, HFnu, and LF/HF did not differ within or between the groups. Kingsley et al. [36] did not find any within- or between-group differences in the LnTP, LnLF, LnHF, or LnLF/HF. Heart rate did not change after resistance exercise [36]. Interventional effects for pain were reported in two studies [35,36]. Gavi et al. [35] demonstrated an improvement in pain by strength training despite no significant difference in the control intervention, stretching. Kingsley et al. reported significant improvement in pain (tender point) from 13±3 at baseline to 8±4 after the resistance exercise.

## Discussion

This systematic review aimed to examine whether patients with CMP had autonomic dysregulation following one-time exposure to aerobic and resistance exercise and whether prolonged use of aerobic and resistance exercise as therapeutic intervention is effective in alleviating the patients' autonomic activities. The results of the review demonstrated that patients with CMP had abnormal autonomic responses during and after aerobic and resistance exercises. Sympathetic and parasympathetic activities showed opposite responses in both exercises compared with healthy controls. However, the heart rate and blood pressure did not differ between patients

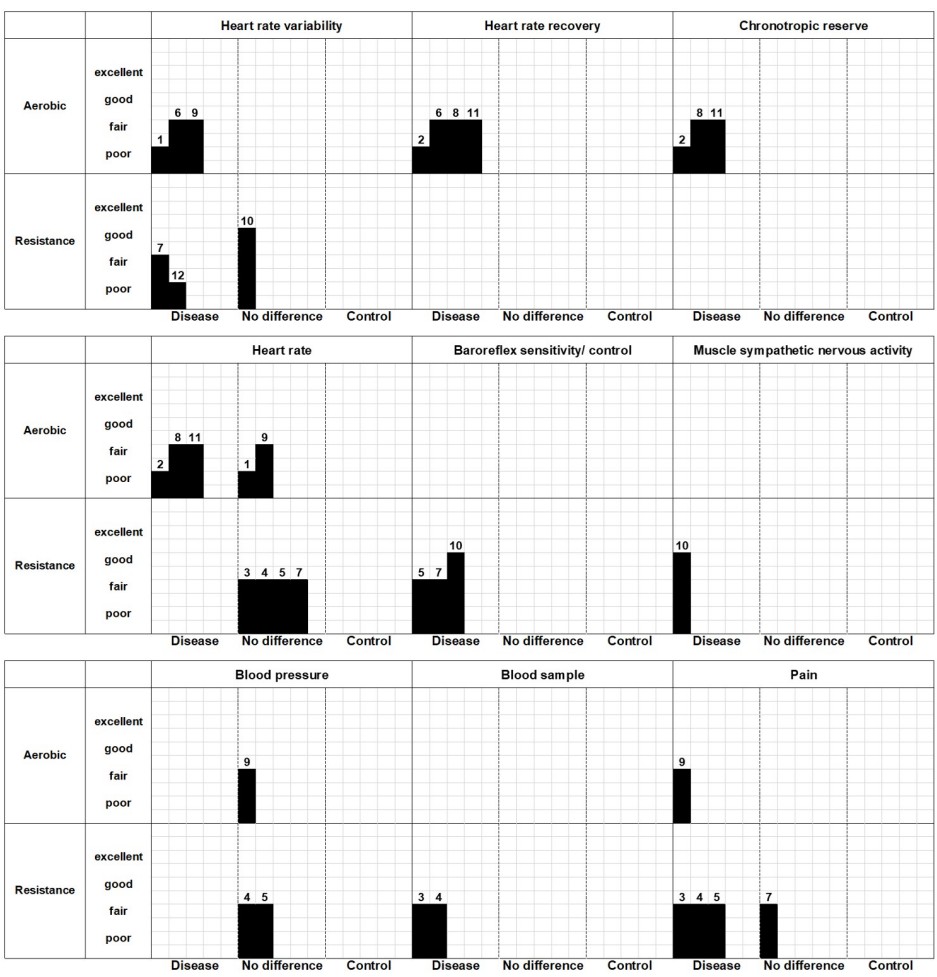

**Fig 3. Harvest plot for observational studies.** This figure shows whether each included article showed abnormal responses in terms of the nine outcome measures. For example, the heart rate variability of the patient group in study No. 1 Cordero et al. [48] showed an abnormal autonomic response compared with healthy controls. The height of each bar indicates the risk of bias as excellent, good, fair, or poor.

and healthy individuals. In interventional studies, some studies revealed positive interventional effects to the autonomic nervous system in patients with CMP, while others did not. Moreover, an association between the reeducation of pain and an improvement was controversial.

## One time exposure to aerobic and resistance exercise

After a single bout of aerobic exercise, dysfunctional sympathetic and parasympathetic responses occur in such patients, in particular, delayed recovery of the autonomic nervous system. Essentially, vagal power is activated, and sympathetic activity is withdrawn at rest, and exercise switches the sympathovagal balance [51,52]. Heart rate recovery can represent blunted sympathovagal balance following exercise [52], and patients with CMP show significantly lower heart rate recovery than healthy controls do [41,43,45,49]. This trend was also demonstrated for the HF of heart rate variability by Oosterwijck et al. [44]. It is reported that disease conditions such as fibromyalgia [53], ankylosing spondylitis [54], and rheumatoid arthritis [55] tend to present with autonomic dysfunctions. Therefore, patients with these disease conditions can show autonomic dysregulation after aerobic exercise, which is expressed as delayed recovery.

**Table 7. Findings of interventional studies.**

| No | Author (Year) | Summary of main findings |
|---|---|---|
| 13 | Bardal (2015) [46] | HRV (SDNN, RMSSD, LF/HF) did not change after the intervention, and did not differ between the groups.<br>HRR did not change after interventions and showed no difference.<br>HR and BP showed no difference after intervention or between groups.<br>Pain (pressure pain threshold) in patients was lower than in controls at baseline (p = 0.01) and post-intervention (p = 0.002), but both groups showed no differences after interventions. |
| 14 | Figueroa (2008) [47] | HRV (TP, RMSSD and HF) increased in patients (p<0.05, <0.05 and = 0.08 respectively), and HRV (LF/HF) showed no change after interventions. |
| 15 | Gavi (2014) [35] | HRV (TP, RMSSD, LF, HF, LFnu, HFnu, LF/HF) showed no differences within and between groups, but only pNN50 at strengthening exercise decreased after interventions (<0.05) with no group difference.<br>Pain (visual analog scale) decreased by both interventions over time (p<0.05) and strengthening exercise was more efficient than stretching (p<0.05). |
| 16 | Kingsley (2010) [36] | HRV (LnTP, LnLF, LnHF, and LnLF/HF) and HR did not show within or between-group differences.<br>Pain (tender points) changed after intervention (<0.05) with group differences. |
| 17 | Sañudo (2015) [37] | HRV (LnTP, LnLF, LnHF, LnRMSSD, HFnu) increased and HRV (LnLF/HF, LFnu, LF/HFnu) after aerobic exercise (p<0.01).<br>HRV (LnHF and LnLF/HF) in patients showed a statistical difference compared with controls (p<0.05).<br>Pain (visual analog scale) did not show any differences within or between groups. |

BP, blood pressure; BRC, baroreflex control; BRS, baroreflex sensitivity; BS, blood samples; CR, chronotropic reserve; HF, high frequency; HR, heart rate; HRR, heart rate recovery; HRV, heart rate variability; LF, low frequency; Ln, natural logarithm; MSNA, muscle sympathetic nervous activity; nu, normalized unit; RMSSD, root mean square of successive differences between normal heartbeats; SDNN, standard deviation of the NN (R-R) intervals; TP, total power.

Resistance exercise also produced different responses in CMP patients than in healthy controls in terms of heart rate variability, baroreflex sensitivity, MSNA, and blood samples. A systematic review [56] stated that healthy participants tended to show parasympathetic withdrawal and sympathetic activation after resistance exercise. The included study [42] showed different autonomic responses, including parasympathetic activation and sympathetic withdrawal. Moreover, lower baroreflex sensitivity in patients was reported in three articles [34,40,42]. Baroreflex sensitivity is a measure of arterial baroreceptor reflex function [57]. Abnormal conditions such as high blood pressure and aging cause decreased baroreflex sensitivity, including in patients with CMP who show dysregulation of baroreflex control during resistance exercise [58]. MSNA, which is a measure of postganglionic sympathetic activity in the upper and lower limbs [59], was assessed in only one study [34]. MSNA normally increases during resistance exercise; however, in the included articles [54], it was higher in patients than in controls, indicating higher activity of the sympathetic nervous system. As prolonged pain activates the sympathetic nervous system through dysfunction of the stress system [60], patients with CMP show higher MSNA. Two articles took blood samples to measure plasma adrenaline, noradrenaline, ACTH, and cortisol [38,39]; only plasma adrenaline showed a lower value in patients than in controls during resistance exercise [38,39]. A previous study reported similar results, in which only an increase in plasma adrenaline with no change in plasma noradrenaline was observed [61]. Plasma adrenaline indicates sympathoadrenal activity [62]; lower plasma adrenaline suggests lower sympathoadrenal activity in the included studies [38,39]. All outcome measures in heart rate variability, baroreflex sensitivity, MSNA, and blood samples could be used to assess dysfunctional responses during and after exercise.

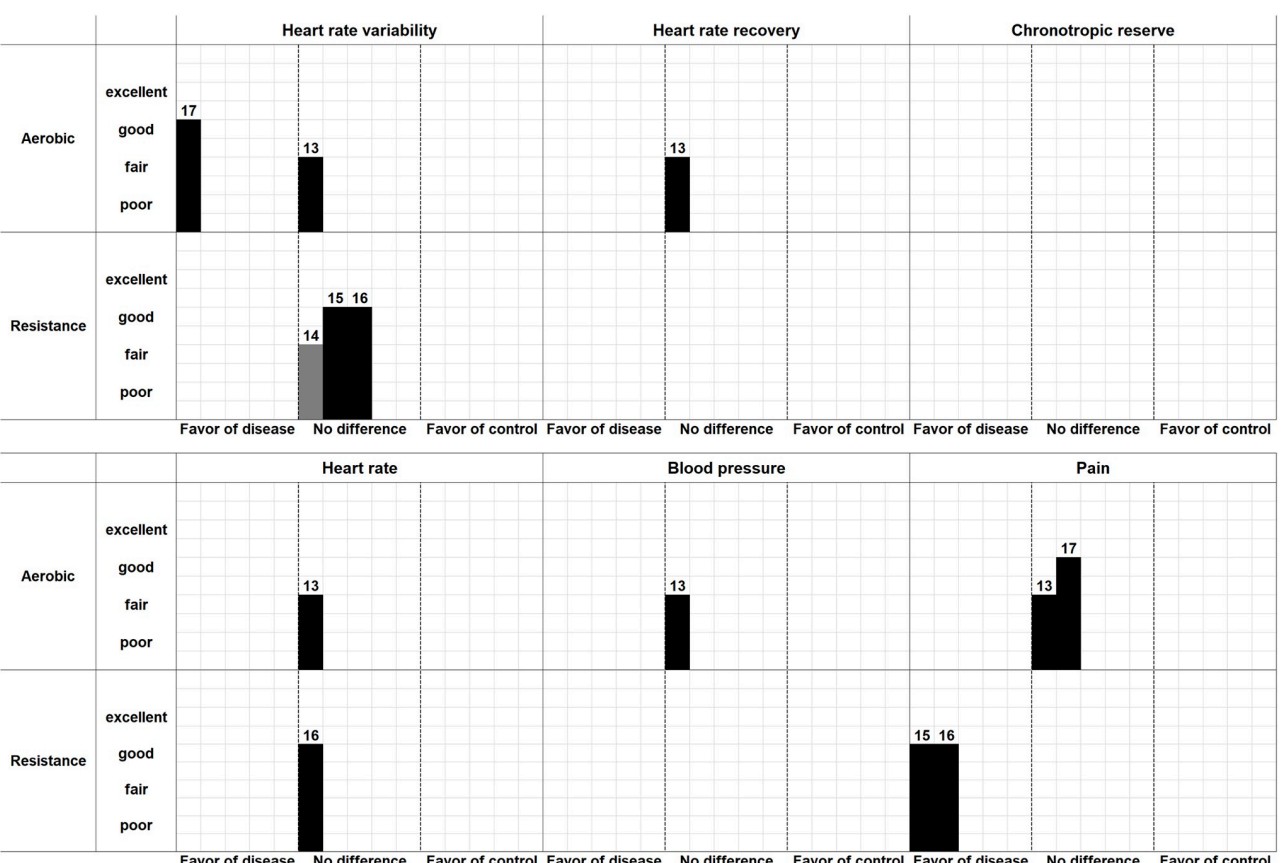

**Fig 4. Harvest plot for interventional studies.** This figure shows that each included article found significant effects in terms of six outcome measures when comparing the patient group with the control group. For example, the HRV of No. 17 Sañudo et al. [37] showed that aerobic exercise was significantly effective in autonomic nervous system (HRV). The height of each bar indicates the risk of bias as either excellent, good, fair, or poor. Although the black bar has two arms, the gray bar has only a single arm–indicating an article had only a patient group without a comparator.

## Aerobic and resistance exercise as therapeutic interventions

Although one study [37] demonstrated the interventional effect of aerobic exercise in improving autonomic dysregulation in patients with CMP, another study [46] did not show an intervention effect. Both the articles utilized the same frequency of exercise at two times per week. However, the duration of interventions of one study [37] was longer for 24 weeks than 12 weeks in another study [46]. Additionally, the intensity of intervention in the study [37] was higher than at 60–80% of the predicted maximum heart rate compared with 50–75% in another study [46]. These studies indicate that higher intensity and longer duration of intervention would be effective in improving the autonomic nervous system function in patients with CMP. In fact, high intensity aerobic exercise has stronger effects in autonomic activities than moderate intensity exercises [63]. Moreover, low-intensity but high frequent interventions could possibily improve the autonomic activities [64], while the two included studies [37,46] had lower frequency at two times per week.

Similar to aerobic exercise, there were discrepancies in the results for resistance exercise. Kingsley et al. [36] did not observe any within-group differences in heart rate variability. Gavi et al. [35] did not report within-group improvements in most of the outcomes of heart rate variability. On the other hand, this study [35] identified that pNN50 improved after resistance

exercise. Figueroa et al. [47] demonstrtaed an interventional effect of heart rate variability in patients with fibromyalgia. The abovementioned discrepancies could be attributed to the duration of interventions. The three studies [35,36,47] were conducted at the same frequency at two times per week. However, a duration of interventions in the articles [35,47] which showed improvement was longer at 16 weeks than 12 weeks in the article [36]. Interestingly, a higher intensity of the interventions could not influence the results because highest intensity such as 75–85% of 1-repetition maximum test [36] did not improve any of the autonomic variables. Higher intensity might affect participants as psychological stress. In healthy individuals, low-intensity resistance exercise has a positive impact on heart rate variability [65], and our results indicated that appropriate duration but not intensity is required for improving autonomic nervous system function in patients with CMP.

An association between a reduction of pain and an improvement in the autonomic nervous system seems to be controversial. In aerobic exercise, Bardal et al. [46] did not report significant improvements in both the pain and autonomic variables. On the other hand, although Sañudo et al. [37] demonstrated an improvement in the autonomic activities, pain did not significantly decrease while a trend of reduction of pain was identified. In resistance exercise, Gavi et al. [35] demonstrated an improvement in both the pain and autonomic activities. However, Kingsley et al. [36] demonstrated a reduction of pain despite no change in the autonomic variables. Furthermore, since Figueroa et al. [47] did not investigate the pain variables, the association between pain and autonomic variables was unclear. Some studies support the reduction of pain to be associated with an improvement in the autonomic variables; however, arriving at a strong conclusion concerning the associations would be challenging owing to insufficient results, such as no significance or pain variables.

## Limitations and future implications

Two limitations affected the strength of our study. First, some articles were assessed as poor by Downs and Black. We included an observational study without any restrictions on the publishing date. Although the Downs and Black checklist can be used for both observational and interventional studies, observational studies tend to have lower scores because some items, such as blinding or population, can be used for interventional studies. In particular, the included studies were essentially in two groups: patients with CMP and healthy controls, and it is difficult to recruit the same population. As the patients we included were relatively rare, this type of grouping was unavoidable. Second, although we searched for and intended to include patients with CMP, the participants of the included studies were limited to fibromyalgia, chronic fatigue syndrome, ankylosing spondylitis, rheumatoid arthritis, and chronic neck and shoulder pain because these diseases tend to present with autonomic dysfunction. Patients with osteoarthritis, chronic low back pain, or any other general musculoskeletal diseases were not included. This could indicate that previous studies did not focus on autonomic dysfunction in patients with musculoskeletal pain. However, recent studies have shed light on autonomic dysregulation in osteoarthritis [17]. The results of this systematic review could promote further research on autonomic dysfunction in patients with CMP, such as those with osteoarthritis.

## Conclusions

Patients with CMP undergoing a one-time exercise challenge demonstrated variable impairments in several indices of autonomic regulation, when compared to healthy controls undergoing a similar exercise challenge. Although therapeutic aerobic and resistance exercise programs extending over periods of weeks to months showed inconsistent impact on the

autonomic indices in patients with CMP, certain trends of improvement in the autonomic dysfunction were identified. On the other hand, an association between a reduction of pain and an improvement in the autonomic variables was controversial. These results would be useful for clinicians and therapists in alleviating chronic pain in patients with CMP, especially fibromyalgia. Further studies of patients with CMP are needed to elucidate whether particular features of an exercise program, including frequency, duration, intensity and type of exercise, are associated with clinically relevant normalization of autonomic regulation and pain. Further investigations of other disease conditions, such as osteoarthritis, are also required to expand the target of the cohort and manage their pain.

## Supporting information

**S1 Table. PRISMA checklist.**
(DOCX)

**S2 Table. Search strategy for MEDLINE.**
(XLSX)

**S3 Table. Precise results of the quality assessments.**
(PDF)

## Acknowledgments

We would like to thank Editage (www.editage.com) for English language editing.

## Author Contributions

**Conceptualization:** Hironobu Uzawa, Shinta Takeuchi, Yusuke Nishida.

**Investigation:** Hironobu Uzawa, Kazuya Akiyama, Hiroto Furuyama.

**Methodology:** Hironobu Uzawa, Kazuya Akiyama, Hiroto Furuyama.

**Validation:** Hironobu Uzawa, Kazuya Akiyama, Hiroto Furuyama.

**Writing – original draft:** Hironobu Uzawa.

**Writing – review & editing:** Hironobu Uzawa, Shinta Takeuchi, Yusuke Nishida.

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
