## [Decision Letter · Decision Letter 0]

15 Nov 2022

PONE-D-22-21106Autonomic responses to aerobic and resistance exercise in patients with chronic musculoskeletal pain: a systematic reviewPLOS ONE

Dear Dr. Uzawa,

Thank you for submitting your manuscript to PLOS ONE. After careful consideration, we feel that it has merit but does not fully meet PLOS ONE’s publication criteria as it currently stands. Therefore, we invite you to submit a revised version of the manuscript that addresses the points raised during the review process.

We look forward to receiving your revised manuscript.

Kind regards,

Matias Noll, Ph.D

Academic Editor

PLOS ONE

Journal Requirements:

Reviewers' comments:

Reviewer's Responses to Questions

**Comments to the Author**

1. Is the manuscript technically sound, and do the data support the conclusions?

Reviewer #1: Partly

Reviewer #2: Partly

2. Has the statistical analysis been performed appropriately and rigorously? 

Reviewer #1: I Don't Know

Reviewer #2: Yes

3. Have the authors made all data underlying the findings in their manuscript fully available?

Reviewer #1: Yes

Reviewer #2: Yes

4. Is the manuscript presented in an intelligible fashion and written in standard English?

Reviewer #1: No

Reviewer #2: Yes

5. Review Comments to the Author

Reviewer #1: Manuscript Number: PONE-D-22-21106

Title: Autonomic responses to aerobic and resistance exercise in patients with chronic

musculoskeletal pain: a systematic review

Summary

I appreciate the opportunity to review this interesting report. This is a study to evaluate acute autonomic responses after aerobic and resistance exercises and the interventional effects of both exercises on autonomic dysregulation in patients with chronic musculoskeletal pain (CMP).

1. I think the objective of a systematic review must be specific. Introduction needs more specific statements of the question the review addresses. I think it would be better to condense the sentences. Line 83-91 in page 5, needs to be condensed.

2. In the method section, I think it will be more important to select one of the main outcome among various outcome measures for the autonomic nervous system. Some of the text may be eliminated by the use of Table 2. Table3 can be stated as an Appendix.

3. Some of the included studies and references are out of date.

4. The first paragraph of discussion prefers to summarize the study findings.

5. In conclusion, I would prefer the results must have external validity or generalizability and future study planning.

Minor comments

1. Line 1 in page 5, It would be better to display the references separately.

Reviewer #2: The authors have conducted a systematic review of autonomic responses to aerobic and resistance exercise in patients with chronic musculoskeletal pain. This is an important topic and a difficult one to analyze. There is enormous heterogeneity in the available publications, in terms of study designs, composition of patient groups, interventions, and choice among derived autonomic measures.

The writing style is generally clear and the description of their approach to data retrieval, extraction and analysis is very transparent.

To this reviewer, it would make it even more clear to change the language around when one-time exposure to aerobic or resistance exercise is used to probe autonomic responses versus prolonged use of aerobic or resistance exercise over many months as a therapeutic intervention.

The authors make a strong effort to analyze the impact of exercise as a therapeutic intervention, and noted that some studies showed improvements in certain measures, while others did not. I would have liked to see more analysis about reasons for success or failure of a therapeutic intervention study. Can the authors find any indication of a dose/intensity/duration effect?

As a thought experiment, it could it turn out that :

treatment A

30 minutes of moderate aerobic exercise 3 times weekly for 12 weeks is too "small of a dose" to normalize autonomic responses, while

treatment B.

1 hour of intense aerobic exercise 5 times weekly and 1 hour of intensive resistance exercise 4 times weekly, both for 12 weeks, does produce clinically significant improvements in derived measures of autonomic function.

In many exercise therapeutic intervention studies, the primary aim is to improve function and pain, and normalizing autonomic physiology can be a secondary outcome. It would be worth knowing the extent to which improvement in function or pain was or was not associated with improvement in derived autonomic measures.

In my view, the Conclusion should also make a greater distinction between use of exercise as a probe of autonomic function versus repeated exercise as a therapeutic intervention. I also think that the Conclusion should express greater uncertainty regarding the therapeutic impact of an exercise program. Some readers will read the Conclusion as "exercise does nothing to normalize autonomic regulation" rather than "available studies are mixed and inconclusive regarding impact of exercise on autonomic regulation".

Perhaps something like the following:

Patients with CMP undergoing a one-time exercise challenge showed variable impairments in several indices of autonomic regulation, when compared to healthy controls undergoing a similar exercise challenge.

Therapeutic aerobic or resistance exercise programs extending over periods of weeks to months showed inconsistent impact on autonomic indices in patients with CMP. Further studies of patients with CMP are needed to elucidate whether particular features of an exercise program, including frequency, duration, intensity and type of exercise, are associated with clinically relevant normalization of autonomic regulation.

6. PLOS authors have the option to publish the peer review history of their article (what does this mean?). If published, this will include your full peer review and any attached files.

Reviewer #1: No

Reviewer #2: No

---

## [Author Response · Author response to Decision Letter 0]

24 Dec 2022

Reviewer 1: I have incorporated all of your suggestions into my revision. They were very helpful. Thank you.

Reviewer 2: I have incorporated all of your suggestions into my revision. Thank you for your help.

---

## [Editor Report · Decision Letter 1]

2 Aug 2023

Autonomic responses to aerobic and resistance exercise in patients with chronic musculoskeletal pain: a systematic review

PONE-D-22-21106R1

Dear Dr. Uzawa,

We’re pleased to inform you that your manuscript has been judged scientifically suitable for publication and will be formally accepted for publication once it meets all outstanding technical requirements.

Kind regards,

Andrea Martinuzzi

Academic Editor

PLOS ONE
---

## [Editor Report · Acceptance letter]

4 Aug 2023

PONE-D-22-21106R1 

Autonomic responses to aerobic and resistance exercise in patients with chronic musculoskeletal pain: a systematic review 

Dear Dr. Uzawa:

I'm pleased to inform you that your manuscript has been deemed suitable for publication in PLOS ONE. Congratulations! Your manuscript is now with our production department. 

Kind regards, 

on behalf of

Dr. Andrea Martinuzzi 

Academic Editor

PLOS ONE